# Community Pharmacists’ Knowledge, Willingness, and Readiness to Prescribe Oral Contraceptives in Saudi Arabia

**DOI:** 10.3390/healthcare10030503

**Published:** 2022-03-09

**Authors:** Haya M. Almalag, Wael H. Mansy, Abdulrahman M. Alwhaibi, Wajid Syed, Salmeen D. Babelghaith, Mohamed N. Al Arifi

**Affiliations:** Department of Clinical Pharmacy, College of Pharmacy, King Saud University, Riyadh 12372, Saudi Arabia; halamalq@ksu.edu.sa (H.M.A.); aalwhaibi@ksu.edu.sa (A.M.A.); wali@ksu.edu.sa (W.S.); sbabelghaith@ksu.edu.sa (S.D.B.); malarifi@ksu.edu.sa (M.N.A.A.)

**Keywords:** oral contraceptives, willingness, community pharmacists, knowledge, Saudi Arabia

## Abstract

Background: The role of community pharmacists (CPs) in various healthcare settings is well documented in the literature including providing safe and easy access to medications. Oral contraceptives (OCPs) are the most frequently used method of terminating unwanted pregnancies worldwide. Objective: This study aims to evaluate the Community pharmacist’s knowledge, willingness, and readiness to prescribe OCPs in Saudi communities in Saudi Arabia. Methods: This is across sectional, self-administered questionnaire-based study conducted between May and November 2021 in the central region of Saudi Arabia. The results were presented as frequencies and percentages. Chi-square tests were used to sort significant association between groups. Results: Out of 368 CPs who got the questionnaire, 347 completed (94.3%). Most of CPs were be-tween the ages of 25 and 35, with 76.9% working in chain pharmacies. Of the surveyed CPs, 45.5% had >24 months of experience in community pharmacies. 41.8% of them prescribed more than six prescriptions for OCPs/week. The patients’ safety (77.2%), physician’s resistance (54.5%), and CPs objection based on religious purposes (36.9%) and lack of time (29.7%) were the most commonly cited barriers among CPs. CPs who worked in chain pharmacies were significantly too busy (*p* = 0.038) to prescribe OCPs. Also, community pharmacists with experience of more than two years significantly agreed not to prescribe OCPs due to religious convictions (*p* = 0.009). Conclusion: The current study revealed that most of the CPs were knowledgeable about OCPs. Additionally, most of them were likely to prescribe oral contraceptives. We further suggest overcoming the barriers associated with contraceptives among CPs and providing sufficient training to improve the oral contraceptive prescriptions in CPs is needed.

## 1. Introduction

Unwanted pregnancies are becoming increasingly more common these days, with the worldwide prevalence of 45% [1,2,3]. It has been proven that having a child in early adolescence is linked to poor maternal health and has an impact on the patient’s social and economic position [4]. Despite the increased usage of contraceptive methods to prevent undesired births, 95% of prior reports found that poor contraceptive use was one of the contributing factors [5]. OCPs assist families with accomplishing their desired number of children and determining the spacing between pregnancies, as well as with terminating undesirable pregnancies according to healthcare specialists’ instructions [2,3].

OCPs are among the most common birth control methods in the United States (US), mainly used by young and middle-aged women. OCPs are high-priority tablets to prevent unexpected births due to their hassle-free availability at pharmacies [6]. In addition, prior research has shown that having access to pharmacies would improve or reduce the incidence of pregnancies though contraceptive prescription [7]. The importance of community pharmacists (CPs) in the healthcare system has been documented worldwide [8,9,10]. CPs are frontline health care experts that give patients and the general public trustworthy and evidence-based health information without requiring consultation or an appointment [8,9]. In addition, research has indicated that CPs aid patients with medication counseling, thereby improving their quality of life [9].

According to previous reports, early age at the time of menarche and late marriage are the underlying factors contributing to premarital sex and unintended pregnancies [10]. Other studies reported poor social support, smoking, and alcoholism are some of the other factors that contribute to unexpected pregnancies [10,11]. Providing an accurate and complete guidance for the use of OCPs is a key social change approach for ensuring the independence of women of childbearing age [12]. Furthermore, patient education and counseling regarding the use of medications, including contraceptives, would improve health outcomes, and this was the primary job of CPs following the consultation [12]. Use of OCPs minimizes unintended pregnancies, allowing populations to be controlled [13]. According to recent estimates Saudi Arabia has risen to fifth place as the most populous country in the Middle East and has seen a significant increase in family planning use, which has been linked to a greater understanding of family planning methods [13,14]. With regards to CPs prescribing OC, most of the earlier literature suggested that women of reproductive age were more comfortable obtaining contraceptives from CPs [15,16]. In Saudi Arabia, the Saudi Food and Drug Authority (SFDA) is solely responsible for the regulation of the drug market; practicing pharmacists and other health care professionals are under the control of the Saudi Commission for Health Specialties [17].

The involvement of CPs in prescribing OCPs is critical, and their knowledge, willingness, and readiness to prescribe oral contraceptives can improve the prescription pattern and end undesirable pregnancies. Despite the wealth of literature about OCPs, there is a scarcity of research on the knowledge, willingness, and readiness to prescribe oral contraceptives from the community pharmacists’ perspective in Saudi Arabia. Therefore, this study aimed at evaluating the community pharmacists’ knowledge, willingness, and readiness to prescribe OCPs in Riyadh, Saudi Arabia.

## 2. Methods

### 2.1. Study Design

This was a cross-sectional self-administered questionnaire-based study conducted over seven months between May and November 2021 in the center of Saudi Arabia to evaluate the CPs’ knowledge, willingness, and readiness to prescribe OCPs in Saudi communities.

### 2.2. Sampling

Based on the geographical areas of Riyadh, the sample of registered community pharmacists was determined previously to be 2000 CPs [18,19]. The sample size was calculated using the Raosoft online calculator (www.raosoft.com, accessed on 27 February 2022) An estimated 323 participants were required to achieve a 95% confidence interval level and an accepted margin of error of 5% considering a 50% response distribution. We selected pharmacies from different geographical locations in Riyadh for this study, including the Northern, Southern, Western, Eastern, and Central suburbs. The results of this survey enabled us to assess the willingness and readiness of pharmacists to prescribe OCPs in their communities.

### 2.3. Questionnaire

The questionnaire was developed by reviewing the available literature on the topic [20,21,22]. It consisted of three sections. The first section presented demographic data: age, education, years of experience, years of working at community pharmacies in Saudi Arabia, country of graduation. In addition, this section included some information about the availability of emergency contraceptives at the pharmacy and the number of OCP prescriptions per week. The second section of the survey was adapted from a previous study to determine the pharmacists’ knowledge about the appropriate practices of prescribing OCPs [22]. The data were collected through personal visits of team researchers (one researcher and two Pharm D interns). They asked the participants to select all possible correct answers for each question in this section. This section assessed the participants’ knowledge of oral contraceptives, including their indications, adverse effects, potential drug–drug interactions, and contraindications. The third section evaluated the reasons for willingness to prescribe OCPs and the barriers that may potentially influence community pharmacists’ decisions to prescribe oral contraceptives. It also evaluates the proper use and dosage of OCPs as well as dealing with missing doses. A five-point Likert scale (from “Strongly agree” to “Strongly disagree”) and multiple choices were used in this section when appropriate.

A pilot study was conducted among randomly selected students of a small sample size (*n* = 10) to guarantee reliability of the questionnaire and ease of administration. Cronbach’s alpha score was 0.74 for knowledge questions, while Cronbach’s alpha score for the questions regarding the reasons and barriers to prescribing OCPs was 0.7, which suggests that the questionnaire was reliable for conducting the study.

### 2.4. Data Analysis

The results were presented as frequencies and percentages. The Statistical Package for Social Sciences (SPSS) version 26.0 was used to enter and analyze the data (IBM Corp, Armonk, NY, USA). For qualitative variables, chi-squared tests were used. Significance was defined as a *p*-value of less than 0.05.

## 3. Results

Out of the 368 CPs who received the questionnaire, 347 completed it (94.3% response rate). The majority of the CPs were between the ages of 25 and 35, with 267 (76.9%) working at chain pharmacies and 85.6% (297) working at community pharmacies. Of the surveyed CPs, 45.5% (189) had >24 months of experience at independent community pharmacies, while 29.7% (103) had between 5 and 10 years of practical experience at community pharmacies. Furthermore, slightly more than half of the CPs (55.6%) (193) graduated from Saudi universities. About 41.8% (145) of the surveyed CPs said they prescribed more than six prescriptions for OCPs every week, while 23.3% (81) said they prescribed from four to six prescriptions per week (Figure 1). The vast majority of the CPs (88.2%, or 306) received oral contraceptive education. Furthermore, a detailed description of the demographics and other characteristics of the surveyed CPs are presented in Table 1.

In this study, the majority of the participants (309 (89%)) agreed that OCPs prevent pregnancies, while one-third of them believed OCPs help in the management of acne. With regard to side effects, slightly more than half, 56.8%, cited irregular bleeding, followed by weight gain (76.7%) and mood swings (74.1%). In terms of contraindications, 68.3% indicated breast cancer; most of the participants agreed that OCPs interact with antiepileptic drugs (61.4%). The detailed responses regarding the CPs’ knowledge items of OCPs are given in Table 2.

In terms of proper OCP administration, the majority of the CPs agreed with administering for 21 days in the 5 days of the cycle, which was extended to 7 days. In the event of a missed dose, 65.7% of the CPs recommended taking a new tablet and taking extra care the following week. Furthermore, the majority of the CPs (82.7%) advised taking progesterone tablets at the same time (Table 3). Table 4 shows the knowledge score of the CPs and their various demographic characteristics. The median score differed substantially by pharmacy type (*p* = 0.013), years of working experience (those with >24 months of experience scored 3 (1), whereas those with 6 months and 6–24 months of experience scored 2 (1) (*p* = 0.003)). Similarly, there was a link between the median knowledge score and the number of years spent practicing.

To determine the reasons why they are suited to prescribe oral contraceptives, pharmacists who were interested in changing their prescription practices were asked. The majority of CPs reported that the most common reasons are improved access and advice that will benefit patients (83.6%), followed by oral contraceptives are a significant public health concern (78.8%), and Pharmacists’ professional development (76.4%). A detailed description of the CPs willingness to prescribe oral contraceptives was given in Table 5. Patient safety (77.2%), increased workload 45.8%, lack of training about OCPs (44.1%), Physician resistance 54.5%, reduced trust among CPs to prescribe OCPs 38.9%, and religion prospective 36.9%. (Table 5).

To understand whether demographic data of the CPs impact their response to reasons for the community pharmacists’ willingness to prescribe oral contraceptives, a chi-squared analysis was conducted and revealed a significant association between the education of the CPs and their belief in the patients’ compliance if OCPs were available at community pharmacies (73.5% (225 of 306) of the educated CPs agreed, while only 43.8% (14 of 32) of those CPs who were not educated about OCPs agreed on that). In addition, the majority of CPs believed that OCPs are a significant public health concern. This showed statistical significance concerning years of the CPs working in the Saudi community (*p* = 0.001) and willingness and readiness to prescribe OCPs as presented in Table 5 (those who had more than 20 years of working in Saudi Arabia were willing and ready to prescribe OCPs).

This study revealed the barriers to prescribing OCPs by CPs in their pharmacy which included concerns regarding patients’ safety (77.2%), physician’s resistance (54.5%), and CPs objection based on religious purposes (36.9%) and lack of time (29.7%). A significant association related to practice site was found where CPs who were working in chain pharmacy agreed on pharmacists being too busy to prescribe OCPs(*p* = 0.038). Also, a statistical significance concerning working years of CPs was found where CPs working > 2 years agreed on not prescribing OCPs based on religious beliefs (*p* = 0.009), as presented in Table 6.

## 4. Discussion

The CPs were trained health care professionals and clinically knowledgeable to provide OCPs at their practice site [23,24,25]. Although the number of prescriptions varied depending on the type of pharmacy and practicing pharmacist in community settings. This study highlighted the prevalence of OCPs prescriptions and knowledge of com-munity pharmacists towards OCPs, among Arabic populations which may assist to expand the role of CPs in health care. In this study, half of the CPs prescribed oral contraceptives. These results were comparable to a previous study by Parsons et al. among British community pharmacists, who reported 45.5% of the OCPs prescriptions [23]. Another survey of North Carolina CPs found that 83% of them were likely to prescribe hormonal contraceptives [26]. The reality is that pharmacies may give patients counseling and over-the-counter medications as health care professionals. Pharmacists who want to practice in Saudi Arabia must register with the Saudi Arabian health authorities, assisting pharmacists in learning more about drug pharmacotherapy. There has to be a discussion regarding customers’ concerns about pharmacists’ prescriptions of drugs.

In this study, the majority of the CPs reported that OCPs are indicated mainly to prevent pregnancies, to treat acne. Similar results were found in a previous study by Ibrahim and Hussein in 2017, who reported 74% of the CPs agreed that OCPs are indicated to prevent pregnancies followed by acne, to relieve menstrual cramps [27] (Ibrahim and Hussein, 2017). In the current study, the most cited side effects of the use of OCPs by CPs were weight gain, mood swings, and irregular bleeding. These findings were comparable to previous studies published in other countries [27,28]. For instance, Ibrahim and Hussein et al. reported weight gain, bleeding, and headaches [27]. Similarly, another recent study by Shakya et al. (2020) reported such side effects as irregular menstruation (%), followed by vaginal bleeding (35.9%), nausea/vomiting (35.2%), infertility (28.1%), and headache (18.8%) [29].

In this study majority of CPs (83.6%) agreed improving access and providing benefit to patients were the main reasons for willingness to prescribe oral contraceptives, followed by a significant public health problem (78.7%), to strengthen the relationship with a physician (73.8%) followed by the pharmacist were qualified and educated to prescribe OCPs (74%). Our study findings were similar to a previous study among American pharmacists by Rafie et al. in 2021 who reported, enjoying individual patient contact (94%), followed by professional development (88%), to benefit patients (86%), valuable service (86%), and important public health issue (83%) and strengthen relationships with physicians (52%) [25].

Slightly more than half of the CPs agreed that increasing the financial incentive is another potential reason for OCPs prescription in community pharmacies in Saudi Arabia. This finding is quite the opposite of the previous result by Rafie et al., who found that most of the surveyed CPs Pharmacists were less likely to agree that pharmacists prescribing contraception would increase business or revenue for the pharmacy [25]. Although previous reports revealed that community pharmacies are not the most significant source of the business, the counter products and refilling are at community pharmacies make an actual loss in their original revenue [30,31,32].

According to current findings, the most common barriers to prescribing OCPs among CPS were patient safety (77.2%) followed by increased workload (45.8%), physician’s resistance, and religious objection. Similar results were reported by earlier studies around the world [25,26,27,28,29]. For instance, Seamon et al. reported added responsibility and liability (69.8%) and time constraints (67.2%) [26]. Similarly, Rafie et al. dis-covered that safety issues (76%), contraception-related complications and adverse effects (75%), lack of access to medical records (67%), appropriate self-administration of selected contraceptive method (63 percent), and lack of payment for patient-care ser-vices (63%) were the most common concerns (63%) payment issues (60%), time limits for pharmacists (57%), and greater responsibility and liability concerns (60%) (54%) [25]. According to Rodriguez et al., 2020, an additional barrier for CPs is the need for an appointment and the expense of services [25]. Financial obstacles are also a factor in OCP purchases [33]. As a result, it is widely recognized that identifying existing obstacles is crucial to the successful introduction of OCPs in community pharmacies. A training program to increase prescribing competency and to support CPs, continuing discussions with major buyers in the state for reimbursement purposes, and the development of protocols that allow pharmacists to quickly identify patients who are candidates for prescribing and those who warrant a referral, as well as address potential safety concerns, are all potential solutions to the current identified barriers [25,26].

There are certain limitations to the current study. First, the findings were based on a self-administered questionnaire, which could have increased the risk of biases such social desirability bias or recall bias. Second, the findings were based on a specific region in Saudi Arabia, making them non-representative of other regions and therefore not internationally applicable. Third, because women are still not appointed or allowed to serve in community settings in Saudi Arabia, the study did not include female CPs. Despite these limitations, our research proposes that more emphasis be placed on raising individual awareness of the health care services provided by CPs in order to improve the health of everyone else in the community.

## 5. Conclusions

The current study highlights the role of the community pharmacist in prescribing contraceptives. Most of the CPs were knowledgeable about OCPs additionally the ma-jority of them were likely to prescribe oral contraceptives. Patient safety, physician’s resistance and increased work load were the most commonly cited barriers among Saudi community pharmacists. We further suggest overcoming the barriers associated with contraceptives prescribing in community pharmacy would help in improving the OCPs prescribing. Furthermore, additional training and educational programs concerning the safe use of contraceptive among patients would be helpful to achieve adequate health outcomes.

## Figures and Tables

**Figure 1 healthcare-10-00503-f001:**
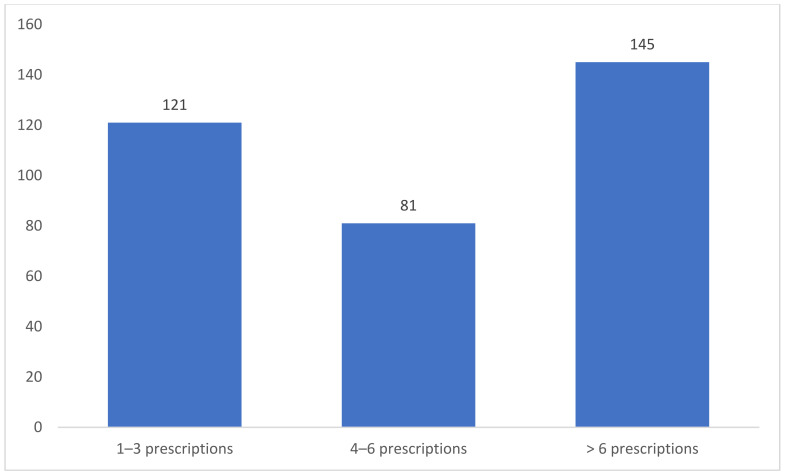
Number of oral contraceptives prescribed or sold weekly.

**Table 1 healthcare-10-00503-t001:** Demographic data of the community pharmacists (*n* = 347).

Variables	*n* (%)
Age (years)	
25–35	267 (76.9)
36–45	68 (19.6)
46–55	11 (3.2)
56–65	1 (0.3)
Work setting	
Chain CPs	297 (85.6)
Independent CPs	50 (14.4)
Years of experience	
<6 months	59 (17.0)
6–24 months	99 (28.5)
>24 months	189 (45.5)
Years of practice as a community pharmacist	
<1	80 (23.1)
from 1 to <5	90 (25.9)
from 5 to <10	103 (29.7)
from 10 to <20	65 (18.7)
>20	9 (2.6)
Country of graduation	
Saudi Arabia	193 (55.6)
Egypt	109 (31.4)
Other	39 (11.2)
Number of oral contraceptives prescribed or sold weekly	
1–3 prescriptions	121 (34.9)
4–6 prescriptions	81 (23.3)
>6 prescriptions	145 (41.8)
Education about oral contraceptives *	
Yes	306 (88.2)
No	32 (9.2)

* Missing data.

**Table 2 healthcare-10-00503-t002:** The general knowledge of the CPs regarding oral contraceptives.

Variables	*n* (%)
Indications of tableted oral contraceptives	
Pregnancy prevention	309 (89.0)
Acne	109 (31.4)
Hirsutism	73 (21.0)
Menstrual cramps	79 (22.8)
Anemia due to iron deficiency	24 (6.9)
Side effects	
Irregular bleeding	197 (56.8)
Nausea	172 (49.6)
Weight gain	266 (76.7)
Mood swings	257 (74.1)
Tenderness of the breast	146 (42.1)
Headache	169 (48.7)
Contraindications	
Migraine	98 (28.2)
Liver tumor	32.6 (113)
Breast cancer	237 (68.3)
Deep vein thrombosis	227 (65.4)
Drug–drug interactions	
Antibiotic	177 (51.0)
Antiepileptic	213 (61.4)
I do not know	26 (7.5)
Indications of progestin-only tablets	
Breastfeeding	250 (72.0)
Active liver	54 (15.6)
Breast cancer within 5 years	102 (29.4)

**Table 3 healthcare-10-00503-t003:** Pharmacists’ knowledge about proper use and missed doses.

Question	*n* (%)
Starting day	
Day 1–5 of the cycle, continued for 21 days, followed by a 7-day tablet-free interval	243 (70.0)
Day 1–7 of the cycle, continued for 21 days, followed by a 7-day tablet-free interval	79 (22.8)
Day 1–10 of the cycle, continued for 21 days, followed by a 7-day tablet-free interval	6 (1.7)
I do not know	19 (5.5)
Missed dose in the first week	
Take a new tablet and use additional precautions for the next week	228 (65.7)
Do not take a new tablet, only use additional precautions throughout the next week	94 (27.1)
I do not know	25 (7.2)
Missed dose in the second and third weeks	
Take a new tablet and use additional precautions and omit the tablet-free interval	101 (29.1)
Take a new tablet and use additional precautions throughout the next week	126 (36.3)
Do not take a new tablet, only use additional precautions throughout the next week	86 (24.8)
I do not know	34 (9.8)
Take only progesterone tablets	
Daily at the same time	287 (82.7)
Any time	42 (12.1)
I do not know	18 (5.2)

**Table 4 healthcare-10-00503-t004:** Community pharmacists’ score of knowledge regarding proper use and missed doses.

Variables	Median (IQR)	*p*-Value
Age groups		0.481
25–35	3 (1)	
36–45	2 (1)	
46–55	2 (1)	
Type of pharmacy		0.013
Chain	3 (1)	
Independent	2 (2)	
Years working at community pharmacies in Saudi Arabia		0.003
<6 months	2 (2)	
6–24 months	2 (1)	
>24 months	3 (1)	
Years of practice as a community pharmacist		0.001
<1	2 (2)	
from 1 to <5	2 (1)	
from 5 to <10	3 (1)	
from 10 to <20	3 (1)	
>20	2 (1)	
Country of graduation		0.052
Kingdom of Saudi Arabia	2 (1)	
Egypt	3 (1)	
Other	2 (1)	
Education about oral contraceptives		0.205
Yes	3 (1)	
No	2 (2)	

**Table 5 healthcare-10-00503-t005:** Reasons for the community pharmacists’ willingness to prescribe oral contraceptives.

Reasons	Agree *n* (%)	Neutral *n* (%)	Disagree *n* (%)
Improved access and advice will benefit patients	290 (83.6)	43 (12.4)	14 (4.0)
Pharmacy staff are well-qualified and well-educated to prescribe and counsel on oral contraceptives	256 (73.8)	65 (18.7)	25 (7.2)
If OCP tablets are available in pharmacies, they may increase compliance *	243 (70.0)	86 (24.8)	18 (5.2)
Pharmacists’ professional development	265 (76.4)	66 (19.0)	16 (4.6)
A significant public health concern **	273 (78.7)	58 (16.7)	16 (4.6)
Relationships with local physicians or clinics should be strengthened	256 (73.8)	68 (19.6)	23 (6.6)
Increased business/revenue	195 (56.2)	106 (30.5)	23 (6.6)

* Significant difference with education about OCPs (*p* = 0.003). ** Significant difference with years working at a community pharmacy in Saudi Arabia (0.001).

**Table 6 healthcare-10-00503-t006:** Barriers that prevent CPs from prescribing oral contraceptive.

Reasons	Agree *n* (%)	Neutral *n* (%)	Disagree *n* (%)
Pharmacists are busy to prescribe oral contraceptives *	103 (29.7)	89 (25.60)	155 (44.7)
Increased workload	159 (45.8)	85 (24.5)	102 (29.4)
Patient safety is a major concern	268 (77.2)	51 (14.7)	28 (8.1)
Insufficient training	153 (44.1)	96 (27.7)	98 (28.2)
Reduced trust in pharmacist competency to prescribe oral contraceptives	135 (38.9)	81 (23.3)	131 (37.8)
Physician’s resistance	189 (54.5)	111 (32.0)	47 (13.5)
Objections based on religion **	128 (36.9)	112 (32.3)	47 (13.5)

* Significant difference with the practice site (0.038) ** Years working at a community pharmacy in Saudi Arabia (*p* = 0.009).

## Data Availability

The datasets used and analyzed during the current study are available from the corresponding author on reasonable request.

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
