# Peer review of "Community Pharmacists’ Knowledge, Willingness, and Readiness to Prescribe Oral Contraceptives in Saudi Arabia"

_healthcare, 2022, doi:10.3390/healthcare10030503_

Round 1

Reviewer 1 Report

The present manuscript evaluated the community pharmacist's knowledge, willingness, and readiness to prescribe OCPs in Saudi communities in Saudi Arabia. The manuscript is structured well. However it lacks to provide significance of the present work to the readers. Hence, the manuscript can be improvised with the following comments.

  1. Introduction section is very short and also it has no discussions on the rationale behind the present study. The authors should include the specifics about the development of the objectives of this study.
  2. 368 CPs were taken for the study and this number was determined by whom? Which of the statistical measures were taken to arrive this number? Pl. discuss
  3. A complete sample questionnaire (without responses) should be provided in the revised manuscript. Also the language of the questionnaire need to be mentioned. 
  4. How the authors do ensure that the questionnaires were filled out by CPs and not by any other labor in the pharmacies?
  5. Data analysis need to be significantly improved in the revision. Only the chi-square analysis is not suffice to arrive a decision on the significant difference among the groups. Additional appropriate statistical test(s) should be assessed to ensure their statistical significance.
  6. Also, why do the authors fix C.I of 95% with p-value of 0.05 why not it could be 99% and 0.01?
  7. The analysed data should be presented with the aid of figures or charts for better visualization and to appreciate the statistical measures.
  8. Few typographical errors were seen. For instance, in page 3, line no. 111: Percentage is missing. (About % (145) of the surveyed CPs said they dispensed more than six prescriptions for OCPs every week, while 23.3 % (81) said they dispensed four to six prescriptions per week.)
  9. Also in some instances, references were given in xxxxx 0000 et al. Correct in the revised version. Use the number format for referencing.

Author Response

I want to thank the reviewers for the time and effort they spent reviewing the manuscript and their valuable and insightful comments, which have improved it substantially. We have carefully addressed all the comments in the revised manuscript inked in red.

The corresponding changes made in the revised paper have summarized the responses to the reviewers below.

The present manuscript evaluated the community pharmacist's knowledge, willingness, and readiness to prescribe OCPs in Saudi communities in Saudi Arabia. The manuscript is structured well. However, it lacks to provide significance of the present work to the readers. Hence, the manuscript can be improvised with the following comments.

1. Introduction section is very short and also it has no discussions on the rationale behind the present study. The authors should include the specifics about the development of the objectives of this study.

Response: We appreciate your comment. The significance of the study has been added and has been made clearer in the last paragraph of the introduction section

2. 368 CPs were taken for the study and this number was determined by whom? Which of the statistical measures were taken to arrive this number? Pl. Discuss

Response: We appreciate your comment; we sincere apologies for the confusion. We have added the requested information. Based on the geographical areas of Riyadh, the sample of registered community pharmacists was determined previously to be 2000 CPs [18,19]. The sample size was calculated using Raosoft online calculator (www.raosoft.com). An estimated 323 participants were required to achieve a 95% confidence interval level and accepted margin of error of 5% and considering a 50% response distribution

3. A complete sample questionnaire (without responses) should be provided in the revised manuscript. Also the language of the questionnaire need to be mentioned. 

Response: Sincere apologies, we have clearly added the requested information.

4. How the authors do ensure that the questionnaires were filled out by CPs and not by any other labor in the pharmacies?

Response: We appreciate your comment. For data collection, we have appointed a researcher and fourth-year students to go personally to the pharmacist to take the data; researchers strictly monitored students.

5. Data analysis need to be significantly improved in the revision. Only the chi-square analysis is not sufficing to arrive a decision on the significant difference among the groups. Additional appropriate statistical test(s) should be assessed to ensure their statistical significance.

Response: We appreciate your suggestion. According to our aim, we have carried out an analysis to assess the Community pharmacist's knowledge, willingness, and readiness to prescribe OCPs in Saudi communities in Saudi Arabia. However, we focused to found association between some of the potential variables, like years of experience, type of pharmacy working, reasons for community pharmacists' willingness to prescribe oral contraceptives, barriers, since in most of the earlier studies.

6. Also, why do the authors fix C.I of 95% with p-value of 0.05 why not it could be 99% and 0.01?

Response: A confidence interval is a range around a measurement that conveys how precise the measurement is. The confidence interval tells you more than just the possible range around the estimate. It also means you about how stable the forecast is. A stable estimate is one that would be close to the same value if the survey were repeated. An unstable estimate would vary from one sample to another. Wider confidence intervals concerning the estimate itself indicate instability. On the other hand, narrow confidence intervals about the point estimate tell you that the estimated value is relatively stable; that repeated polls would give approximately the same results.

7. The analyzed data should be presented with the aid of figures or charts for better visualization and to appreciate the statistical measures.

Response: We appreciate your suggestion; I have presented some of variables in the form of graph

8. Few typographical errors were seen. For instance, in page 3, line no. 111: Percentage is missing. (About % (145) of the surveyed CPs said they dispensed more than six prescriptions for OCPs every week, while 23.3 % (81) said they dispensed four to six prescriptions per week.).

Response: Sincere apologies; we have corrected the typo errors throughout the manuscript.

9. Also in some instances, references were given in xxxxx 0000 et al. Correct in the revised version. Use the number format for referencing.

Response: Sincere apologies; we have corrected the typo errors

Reviewer 2 Report

This paper deals with Community pharmacist's knowledge, willingness, and readiness to prescribe oral contraceptives in Saudi Arabia. However, I cannot see the relevance of this study. My suggestions are given bellow, but the authors must to check/improve the manuscript to give it some relevance.

L64-65. Aim of the study, usual is the Last and Separate paragraph of Introduction. Actual aim of study is not at all interesting or relevant. What makes special this study? Which is its novelty character or its special aspects? Why have the author chosen this topic? What differentiate this paper from others in the similar topic. As I already mentioned above, I cannot see the relevance of evaluating the Community pharmacist's knowledge, willingness, and readiness to prescribe OCPs in Saudi communities in Saudi Arabia. 

In the Material and method section, the  calculus of the sample must be described.

Conclusions are poor, bringing no value to the field.

Author Response

I want to thank the reviewers for the time and effort they spent reviewing the manuscript and their valuable and insightful comments, which have improved it substantially. We have carefully addressed all the comments in the revised manuscript inked in red.

The corresponding changes made in the revised paper have summarized the responses to the reviewers below.

This paper deals with Community pharmacists' knowledge, willingness, and readiness to prescribe oral contraceptives in Saudi Arabia. However, I cannot see the relevance of this study. My suggestions are given bellow, but the authors must to check/improve the manuscript to give it some relevance.

L64-65. Aim of the study, usual is the Last and Separate paragraph of Introduction. Actual aim of study is not at all interesting or relevant. What makes special this study? Which is its novelty character or its special aspects? Why has the author chosen this topic? What differentiate this paper from others in a similar topic. As I already mentioned above, I cannot see the relevance of evaluating the Community pharmacist's knowledge, willingness, and readiness to prescribe OCPs in Saudi communities in Saudi Arabia. 

Response: We appreciate your suggestion. Firstly, we aim to evaluate the Community pharmacist's knowledge, willingness, and readiness to prescribe OCPs in the Riyadh region, Saudi Arabia. This study is particular in improving the prescribing pattern of OCPs, which helps in avoiding unexpected pregnancies and helps in achieving good health outcomes. Furthermore, community pharmacists are the front-line health care professionals, available 24 hrs, without any prior appointment, so most individuals consider a pharmacist visit rather than a physician visit to refill the prescription. Therefore, accessing the knowledge and willingness would help improve the prescription pattern also help in enhancing the awareness about the safe use of OCPs in the community.

In the Material and method section, the calculus of the sample must be described.

Response: We appreciate your suggestion. I have included it in the methods section.

Conclusions are poor, bringing no value to the field.

We appreciate your suggestion. I have re-written the conclusion, which appears in red colour.

Reviewer 3 Report

The manuscript titled “Community pharmacist's knowledge, willingness, and readiness to prescribe oral contraceptives in Saudi Arabia” was generally well organized. The authors intended to reveal the role of community pharmacists in prescribing oral contraceptives in Saudi Arabia, from the perspectives of personnel knowledge, willingness, and readiness. The main contribution of the article is sending out the message of enhancing awareness of the health care services provided by community pharmacists, especially under particular circumstances in Saudi Arabia. 

I recommend that this paper be accepted after minor revision. However, there exist several problems with more considerations granted.
- Clarification on “prescribing” is strongly needed. From what I read, the author meant “dispensing” in some places rather than “having the right to prescribe” oral contraceptives. A huge difference between these actions. 
- With this being said, there exist additional questions/introductions/discussions worth mentioning regarding the legislation/laws in Saudi Arabia that may prevent the legal identity of pharmacists as the “prescribers”. 
- Several missing data claimed under the tables needed more explanations in detail. 
- Improvement in formatting tables and paragraphs should be considered for better readability and demonstration. Problems include inconsistent bolding/capitalization, different font sizes, extra space, missing stop signs, etc. A comprehensive readthrough is recommended.  
- Would like to see the questionnaire (structure/content) as part of supplementary materials. 

Author Response

I want to thank the reviewers for the time and effort they spent reviewing the manuscript and their valuable and insightful comments, which have improved it substantially. We have carefully addressed all the comments in the revised manuscript inked in red.

The corresponding changes made in the revised paper have summarized the responses to the reviewers below.

The manuscript titled "Community pharmacist's knowledge, willingness, and readiness to prescribe oral contraceptives in Saudi Arabia" was generally well organized. The authors intended to reveal the role of community pharmacists in prescribing oral contraceptives in Saudi Arabia from the perspectives of personnel knowledge, willingness, and readiness. The main contribution of the article is sending out the message of enhancing awareness of the health care services provided by community pharmacists, especially under particular circumstances in Saudi Arabia. 

I recommend that this paper be accepted after minor revision. However, there exist several problems with more considerations granted.
- Clarification on "prescribing" is strongly needed. From what I read, the author meant "dispensing" in some places rather than "having the right to prescribe" oral contraceptives—a considerable difference between these actions. 

Response: We appreciate your suggestion. I have corrected throughout the manuscript

- With this being said, there exist additional questions/introductions/discussions worth mentioning regarding the legislation/laws in Saudi Arabia that may prevent the legal identity of pharmacists as the "prescribers". 

Response: We appreciate your suggestion. I have added some information about drug regulations and registration. In Saudi Arabia, the Saudi food drug authority (SFDA) is solely responsible for the regulations of the drug market; practising pharmacy other health care professionals were under the control of the Saudi Commission for health specialities. [15].

- Several missing data claimed under the tables needed more explanations in detail. 

Response: We appreciate your comments. Missing data means some of the pharmacists did not answer complete questionnaires. I mean, some of them left 1 or 2 items.

- Improvement in formatting tables and paragraphs should be considered for better readability and demonstration. Problems include inconsistent bolding/capitalization, different font sizes, extra space, missing stop signs, etc. A comprehensive read through is recommended.  

Response: We sincerely apologize. We have revised the whole manuscript

- Would like to see the questionnaire (structure/content) as supplementary materials. 

Response: we have added it in supplementary materials

Reviewer 4 Report

Thanks for the opportunity to review. The paper will need extensive work before publication but it is a worthy subject to publish.

Page 1: Line 10 add settings after healthcare; Line 11 remove in; Line 13 remove main; Line 19 remove it; line 41 change pills to tablets and throughout paper; line 43 changed reduced to reduce; line 43 spell out CP.

Page 2 line 52 change to alcoholism;  line 58 remove Correct; line 68 remove was; line 68 add study after based; line 82 country of graduation

Page 3 line 107 change community to independent, line 111 missing percentage number

Page 4 change pills to tablets in chart; line 128 pill to tablet

Page 5 pills to tablets in charts

Page 6 change were asked to were surveyed; line 140 change to most common reasons; line 151 delete on; line 152 delete on that

Page 7 Table 6 change busily t busy; line 172 delete was; Line 180-184 need to rewrite, not clear; line 186 add the before majority, delete were, line 197 need to rewrite. not clear; line 204 delete to

Page 8 change opposing to opposite of; line 210-213 unclear; line 227 change comfort to competency; line 231 change currently to current; line 233 change to limitation to this study

Author Response

I want to thank the reviewers for the time and effort they spent reviewing the manuscript and their valuable and insightful comments, which have improved it substantially. We have carefully addressed all the comments in the revised manuscript inked in red.

The corresponding changes made in the revised paper have summarized the responses to the reviewers below.

Thanks for the opportunity to review. The paper will need extensive work before publication but it is a worthy subject to publish.

Page 1: Line 10 add settings after healthcare; Line 11 remove in; Line 13 remove main; Line 19 remove it; line 41 change pills to tablets and throughout paper; line 43 changed reduced to reduce; line 43 spell out CP.

Response: We sincerely apologise, we have revised the whole manuscript

Page 2 line 52 change to alcoholism; line 58 remove Correct; line 68 remove was; line 68 add study after based; line 82 country of graduation.

Response: We sincerely apologise, we have corrected the typo errors in the whole manuscript.

Page 3line 107 change community to independent, line 111 missing percentage number.

Response: We sincerely apologise, we have corrected the typo errors in the whole manuscript

Page 4 change pills to tablets in chart; line 128 pill to tablet

Response: We sincerely apologise, we have corrected the type of errors in the whole manuscript

Page 5 pills to tablets in charts

Response: We sincerely apologise, we have corrected the typo errors in the whole manuscript

Page 6 change were asked to were surveyed; line 140 change to most common reasons; line 151 delete on; line 152 delete on that.

Response: We sincerely apologise, we have corrected the typo errors in the whole manuscript

Page 7 Table 6 change busily t busy; line 172 delete was; Line 180-184 need to rewrite, not clear; line 186 add the before majority, delete were, line 197 need to rewrite. not clear; line 204 delete to

Response: We sincerely apologise, we have corrected the typo errors in the whole manuscript

Page 8 change opposing to opposite of; line 210-213 unclear; line 227 change comfort to competency; line 231 change currently to current; line 233 change to limitation to this study

Response: We sincerely apologise, we have corrected the typo errors in the whole manuscript.

Round 2

Reviewer 1 Report

The authors have addressed all the raised queries satisfactorily. Hence it can be considered for the publication in the Healthcare journal 

Reviewer 2 Report

Authors responded to my requests